**Data Availability Statement:** The complete description of the dataset generated during and/or analysed during the current study are available

# On the forms, contributions and impacts of community mobilisation involved with Kerala's COVID-19 response: Perspectives of health staff, Local Self Government institution and community leaders

**Gloria Benny**[1]*, **Hari Sankar D.**[1], **Jaison Joseph**[1], **Surya Surendran**[1], **Devaki Nambiar**[1,2,3]

**1** The George Institute for Global Health, New Delhi, India, **2** Faculty of Medicine, University of New South Wales, Kensington, Australia, **3** Prasanna School of Public Health, Manipal Academy of Higher Education, Manipal, India

* gbenny@georgeinstitute.org.in

## Abstract

### Background

Kerala, a south Indian state, has a long and strong history of mobilisation of people's participation with institutionalised mechanisms as part of decentralisation reforms introduced three decades ago. This history formed the backdrop of the state's COVID-19 response from 2020 onwards. As part of a larger health equity study, we carried out an analysis to understand the contributions of people's participation to the state's COVID-19 response, and what implications this may have for health reform as well as governance more broadly.

### Methods

We employed in-depth interviews with participants from four districts of Kerala between July and October, 2021. Following written informed consent procedures, we carried out interviews of health staff from eight primary health care centres, elected Local Self Government (LSG, or *Panchayat)* representatives, and community leaders. Questions explored primary health care reforms, COVID responses, and populations left behind. Transliterated English transcripts were analysed by four research team members using a thematic analysis approach and ATLAS.ti 9 software. For this paper, we specifically analysed codes and themes related to experiences of community actors and processes for COVID mitigation activities.

### Results

A key feature of the COVID-19 response was the formation of Rapid Response Teams (RRTs), groups of lay community volunteers, who were identified and convened by LSG leaders. In some cases, pre-pandemic 'Arogya sena' (health army) community volunteer groups were merged with RRTs. RRT members were trained and supported by the health

within the supplementary file. However due to ethical restrictions and for data confidentiality the full transcripts from the study cannot be made available.

**Funding:** DN was awarded with Wellcome Trust/ DBT India Alliance Fellowship (https://www. indiaalliance.org) (Grant number IA/CPHI/16/1/ 502653) The funders had no role in study design, data collection and analysis, decision to publish, or preparation of the manuscript.

**Competing interests:** The authors have declared that no competing interests exist.

departments at the local level to distribute medicine and essential items, provided support for transportation to health facilities, and assisted with funerary rites during lockdown and containment period. RRTs often comprised youth cadres of ruling and opposition political parties. Existing community networks like Kudumbashree (Self Help Groups) and field workers from other departments have supported and been supported by RRTs. As pandemic restrictions eased, however, there was concern about the sustainability of this arrangement as well.

## Conclusion

Participatory local governance in Kerala allowed for the creation of invited spaces for community participation in a variety of roles as part of the COVID 19 response, with manifest impact. However, the terms of engagement were not decided by communities, nor were they involved more deeply in planning and organising health policy or services. The sustainability and governance features of such involvement warrant further study.

## Introduction

The COVID-19 pandemic has revealed the criticality of community participation in ensuring healthcare access and service continuity, especially In low resource settings [1, 2]. Even outside of the pandemic context, community participation in health has been associated with improving healthcare access, utilisation [1, 3] and assuring the rights of populations [4]. Within the health context, community participation can be defined as the engagement of individuals in the community having shared interests or identities in planning, organising, and implementation of health promotion, prevention, care, treatment, rehabilitation, and palliation [3, 5]. It is also a "process by which members of the community, either individually or collectively and with varying levels of commitment" developing and assessing their needs, implementing solutions and finding and maintaining organizational support [4].

Kerala, a south Indian state, has a long history of social and political mobilisation and of people led social movements. Even prior to the formation of Kerala state, campaigns and struggles for women empowerment, literacy, against caste discrimination, land and tribal rights have been underway [6, 7]. The state established pro poor reforms which include the building of efficient public distribution systems and land reforms [5]. People's participation mechanisms and decentralisation reforms have been institutionalised since the mid-1990s [8]. Participatory and bottom-up planning not just in health but across sectors were the hallmarks of the People's Planning Campaign (PPC) which was launched to implement India's 73[rd] and 74[th] constitutional amendments mandating local governance and subsidiarity in the state and an enhanced role for Local Self Government (LSG) [5, 9]. Under these reforms, funds, functionaries, and functions were transferred to LSGs [10], strengthened by support and training for local health planning at the LSG level, which also helped deepening community participation for health and allied activities in the initial days of decentralization by streamlining the networks of women Self Help Groups (SHG) [11]. These reforms allowed for many communities and civil society-based initiatives related to health and welfare to be established, adapted and eventually scaled up in the state [12–14].

Kerala has witnessed several community led initiatives in the past 5 years on managing disease outbreaks and disaster management [15, 16]. Previous experiences of handling two

consecutive floods in 2019 and 2020, and Nipah outbreak during 2018 have been made possible through devolution of power to LSGs and a strong involvement of *gramasabhas* (village councils) [17, 18]. The media in particular has highlighted community participation in response to a host of disasters, from floods to Nipah, from COVID-19 to Zika [19, 20]. Kerala state's preparedness to face contingencies like COVID pandemic has been attributed to these prior experiences [21, 22].

Some have specifically recognised the role of community participation and intersectoral convergence therein [23]. Community Health Workers (CHW) have implemented measures to better contain the spread of the pandemic. LSGs have steered the coordination of efforts and activities by volunteers [24]. There is journalistic and anecdotal information suggesting that Kerala's COVID response included formal and informal mechanisms comprising a host of players including Kudumbashree (Kerala's government-run women's SHG program), voluntary organisations, youth associations as well as new formations like the Rapid Response Team (RRT) at the ward level set up during the pandemic [25]. RRTs were constituted formally by the state mechanism to have a supervised and monitored support for the state pandemic mitigation efforts [26]. They were volunteers from the community trained by LSGs and staff at Primary Health Care facilities (PHCFs), under the broader supervision of LSG elected members.

Much global attention has been paid to the impact of CHW programs, with India's ASHA (CHW) receiving a 2021 Global Health Leaders award from the World Health Organisation [27]. However, in states like Kerala, these actors were supported by other actors including LSGs, community structures like members of *Kudumbashree*, and emergent community formations like RRTs. To best of our knowledge, the roles and impacts of pandemic led community participation like RRTs have not been studied systematically, nor is it clear how they specifically interacted with health system actors who were core responders during the pandemic. As part of a larger health equity study, we carried out an analysis to understand the contributions of community participation mechanisms to the state's COVID response. Our analysis explored the provenance of community led initiatives, their functioning, and implications for health reform as well as local health governance more broadly.

## Methods

### Study setting and design

We carried out qualitative in-depth interviews, as part of a larger five-year collaborative health systems research project supporting health reform processes in the state of Kerala, India. Fieldwork for this study was conducted between the months of July and October of 2021. Eight PHCFs were selected using multi stage random sampling (detailed elsewhere [28]) across four districts in the state.

### Participant selection

The recruitment of participants (health system actors) for the study was carried out using purposive criterion sampling–we intended on having representation from PHC staff (clinical and public health), LSG members and community actors. We interviewed three categories of participants from each of the selected PHCFs. 1. Elected representatives of LSG (Panchayat President-PP, Health Standing Committee Chairperson- HSC, Ward member- WM, Welfare Standing Committee Chairperson- WSC). 2. Department of health staff (Medical Officer- MO, Nursing officer-NO, Health Inspector-HI, Junior Health Inspector-JHI, Junior Public Health Nurse-JPHN, Palliative Nurse-PN, ASHAs). 3. We requested participants from first two

categories to nominate a community leader from the area, defined as someone known to be socially committed and actively engaged in the community.

## Data collection process

A team of three researchers (two males, one female) trained in qualitative research methods with postgraduate degrees in areas of public health, social work, and development carried out the fieldwork. Few members of the research team met the respective District Medical Officers and requested permission from them to carry out the study in the specific facility along with providing permission for overall study from department of health- Kerala. We also informed and requested permission from Medical Officers and the Panchayat Presidents of the respective PHCF of study concerned to conduct interviews with the staff from their institution. Selected participants were provided with participant information sheets in both English and Malayalam and were briefed about the study in Malayalam. In-person and online interviews were carried out at the participant's convenient time and in language of preference. In person interviews were carried out within their workplace premise where privacy was maintained. Some participants opted for online interviews due to their unavailability during our field work (some of them had COVID infection, while others were busy with COVID management activities). For them we carried out telephonic interviews. Three participants were not available for interviews during our data collection period. After at least three rounds of interview rescheduling per participant, we removed them from the participant list.

The in-person interviews were conducted at participant workplaces. The interviews began with a detailed written informed consent process, with separate permission for audio-recording. For those interviews held via telephone, the soft copy of the consent form was sent to the participants and were returned with the signatures. A pilot tested semi-structured interview guide was used for carrying out IDIs. The interviews lasted between 20 and 60 minutes. The interviews were held in Malayalam and covered a range of questions including participants' experiences and impressions of recent Aardram reforms in the state, the role(s) and involvement of LSG departments in these reforms, changes and challenges experienced during COVID and how they were addressed, information about the presence of vulnerable populations in the respective panchayats that were felt to be left behind from health and other services, and suggestions from the participant's on how to improve the primary health care in the state. Data saturation began to be achieved before the last third of participants were interviewed, however the criterion sample of participants was covered to ensure representativeness from different health system actors from selected study sites. All audio-recordings of the interviews were uploaded to a secured database, accessible only to the research team. Field notes were created to support transcripts in addition to the transliterated English transcripts of the interviews prepared by the research institution empanelled third party. To ensure quality of the transcripts, each transliterated English transcript was reviewed by one of the three research team members.

## Analysis

Thematic analysis was carried out to analyse the data collected using the ATLAS.ti 9 software [29]. A four-member research team (DN, HS, JJ, GB) was involved in data perusal and inductive theme generation. A draft code book was generated using the data from the inductive coding process. Multiple discussions were held among four-member team resulted in finalizing the codebook and thematic structures. During these meetings, the team came to a consensus on merging some codes with other and redefining the codes for greater clarity. ATLAS.ti 9 files from the four team members were merged and analysis were consolidated in a narrative format.

**Table 1. Participant characteristics.**

| Category | Designation | Female | Male | Total |
|---|---|---|---|---|
| LSG leadership | Panchayat President | 3 | 4 | 7 |
| | Panchayat Vice-President | 0 | 1 | 1 |
| | Health Standing Committee Member | 2 | 5 | 7 |
| | Welfare Standing Committee Member | 1 | 0 | 1 |
| | Ward Member | 0 | 1 | 1 |
| Community leaders | Community Leader | 1 | 6 | 7 |
| Health care providers | Medical Officer | 5 | 3 | 8 |
| | Health Inspector | 1 | 5 | 6 |
| | Public Health Nurse | 4 | 0 | 4 |
| | Junior Health Inspector | 0 | 7 | 7 |
| | Junior Public Health Nurse | 11 | 0 | 11 |
| | Nursing Officer | 3 | 0 | 3 |
| | Palliative Care Nurse | 1 | 0 | 1 |
| | Community Health Worker | 16 | 0 | 16 |
| | Total Participants | 48 | 32 | 80 |

For this paper, we specifically analysed codes and themes related to experiences of community actors and processes in COVID mitigation and response activities. The interview questions pertaining to this analysis and codes emerged that were used in analysis for this paper are attached as Annexure-1. Quotations were indexed in MS Excel, analysed and combined to form themes which were further concise to form the study results of the paper.

### Ethics approval

This study was approved by the Institutional Ethics Committee of the George Institute for Global Health (Project Number 05/2019). Before initiating the project, permissions were obtained from Department of Health and Family Welfare, Government of Kerala.

## Results

We conducted in-depth interviews with 80 participants who were from state health systems, eight PHCFs, elected representatives from Panchayats of these eight primary health facilities (called Family Health Centers), as well as community leaders nominated by the above two categories of participants. There were 56 participants from the health department, 17 from LSG institutions (i.e. *panchayat* members) and seven community leaders (see Table 1).

Our study participants described the various community formations during the period of COVID-19 pandemic, their linkages to system functions and activities in collaboration, and their impact. These are detailed below.

### What community formations became active during the COVID-19 pandemic?

Kerala has a strong legacy of community participation for social causes including health. Various institutional arrangements for community participation in health have been in place in Kerala since at least the time of the PPC in the 1990s. As part of Aardram health sector reforms put in place 2017 onwards, community volunteer groups called *Arogya senas* (literally translated into health armies) were mandated by the health department to assist staff with documenting local health and health related issues, mostly surveillance of vector-borne disease and

outbreaks in general. Prior to COVID pandemic, however, *Arogya sena* formation, activities and presence were erratic: a household survey of over 3,000 individuals we carried out in the same four districts in November of 2019 found that 1.4% of individuals surveyed knew what the *Arogya sena* was. These were early days of reform: health system actors we interviewed and even those we spoke to informally indicated that these groups were taking some time to establish themselves and only in some locations were they active in work related to prevention of communicable disease outbreaks locally.

Early into the COVID-19 pandemic in 2020, however, this changed. The government created state level RRTs and instructed LSG leaders to establish ward-level RRTs. Health workers we interviewed noted that *Arogya senas* constituted as part of Aardram reforms evolved to form RRT, once COVID 19 pandemic started in 2020. A medical officer said the following:

> *Aarogya sena was started 5 years ago. In a locality of 25 houses, somebody who stays there was assigned,. . . If some disease happens there or some communicable diseases occur there, they will come to know. . . At that time (pre-COVID time), this did not work out well. But once COVID came, several people started volunteering. Now we have a really good RRT team. . .They take the initiative for several issues. They work really well as informers and for source reduction (communicable disease control). (41_MO)*

One community leader however mentioned a different perspective regarding the formation of RRTs. This participant pointed to the community participation which followed the 2018 floods in the state as the origin for RRT evolution: *"RRT was formed initially as a disaster management group during the floods. We then diverted the group to COVID response. We could mobilise the RRT as a response to both these crisis, even though the ways to handle a flood and pandemic are completely different."* (41_CL)

Members of established youth organizations affiliated to political parties supplemented and/or supported *Arogya senas*, as observed by a Community Health Worker. In other cases, serving or former government employees were also closely involved. A community leader we interviewed, whose wife is an elected representative of the Panchayat, noted an adaptation they had incorporated in the membership of the RRTs in their ward.

> *I chose 22 members for the RRT [in our ward]. This is more than the number they had suggested. Among them, 18 are men and 4 are women. By women, I mean girls. To [supervise] this group, we also had two teachers. . . They are government employees. They helped us. We publicised four contact numbers as the control room. One was mine, one belonged to one of these teachers, my wife's number and another woman's number. (32_CL)*

## What did community formations do?

RRTs received directions from the Health and Family Welfare Department as well as the LSG department. Weekly trainings by *panchayat* officials were conducted online or in person, in abidance of COVID restrictions. *Panchayat* leaders used instructions provided by their local PHC and identified contextual needs to handhold volunteers. Volunteers were also provided with personal protection equipment in circumstances as needed. This training support given to volunteers was resonated by both a health facility official and by a community leader.

> *. . .during the first phase of COVID, before any others in the (anonymised) district, we met at the Panchayat level. . . we selected 10 volunteers from each ward and prepared a list of almost 160 volunteers. We selected a leader from each ward. . .So we had a working system earlier*

*and later it became a common trend, and it was made official. We had formed the RRT by then. . .There was training at the panchayat level through Zoom.* (11_MO)

*The training they received as per the government instruction made them able to understand the situation at the panchayat level and act accordingly.* (22_CL)

Panchayat leaders provided essential Personal Protective Equipment (PPE) for RRT volunteers so they would avoid risk of COVID-19 infection. One Panchayat leader also commented on this, saying that *"we (Panchayat) bought masks, sanitisers and other peripherals for the volunteers (RRTs) to fulfil their duties."* (22_PP)

There was a wide array of activities that were assigned to RRTs, ranging from support for basic supplies to medicine support, and funerary assistance. As one panchayat leader pointed out, in the initial period, even food access was a challenge. Food made in community kitchens with the support of Kudumbashree SHG members and panchayats was delivered free to people who were confined to their homes either due to lock-down protocols in the local area or due to the quarantine imposed on households where someone had tested COVID positive.

RRTs were involved with support for transportation of patients to health facilities, and in distribution of medicine and essential items for persons unable to get medicines on their own, persons who were ill and bedridden. One Health Standing Committee leader furnished an example:

*If someone's condition worsens and there is an emergency, they [the RRTs] provide ambulance service. . .They deliver food (and medicines) for patients. Sometimes all the medicines would not be available in our pharmacy. They get it from other pharmacies as soon as I call and inform them and deliver it to the patients.* (31_HSC)

In some cases, RRT members were able to carry out tasks that there was nobody else to help with, like funerary arrangements. As explained by a panchayat leader:

*One incident that happened in my ward was the sudden death of a person due to COVID. He only had a brother and unfortunately, none of the family members were able to claim the dead body due to risk of infection. Members of (political party–name masked) were ready to carry his body wearing PPE kits at the time.* (41_PP)

Some volunteers were also trained to provide psycho-social support to families when members or friends had a COVID positive diagnosis. RRTs had a significant role in helping people to register for COVID vaccination by going door-to-door to encourage enrollment. Finally, in some places, initiatives like small surveys were carried out by civil society organisations to support LSG department and Health department activities as well. However, there were criticisms related to political favouritism during COVID vaccination priority listing. A Medical Officer expressed his appreciation that *"many gave access to their buildings to shelter COVID patients. . ., sponsored different things. . .In their respective wards, ward members unified these efforts. With the cooperation from many departments, these activities were done very well. Similarly, many volunteers came forward to manage crowds [at the COVID vaccination centres and general crowd as well] and ensure [the] COVID protocol was followed."* (42_MO)

In one of our study districts, RRTs had been collaborating with complementary community formations like the *Mash* (means teacher) team. This team of government school teachers was deployed by the district authorities to monitor compliance to COVID-19 guidelines at the ground level. Over time, this initiative was scaled up to other districts as well. The MO from

this district also mentioned another ward-level approach for disease control measures called *Jagratha Samithi* (or awareness committee). This committee comprised frontline health workers (called ASHAs), public health staff, creche workers (called Anganwadi workers), elected representatives for the ward level at LSG, and local Kudumbashree members. Along with the Panchayat leadership, both *Mash* and *Jagratha Samithis* oversaw response efforts at the ward level by RRT volunteers and other youth volunteer groups, making household visits as necessary:

> *In [the] ward level we have Jagratha Samithi (health force led by health department and LSG). . . The convener of the Jagratha Samithi is an RRT member and the chairman is the ward member. Then we have a team of people which is only in the (masked the name) district, named as Mash. Other than them, we have 3 or 4 local people and police. So, this is the team known as Jagratha Samithi. Every ward has this kind of team, and they are having regular meetings and work. But I think it should be more efficient.* (12_MO)

A final role that community members and formations came to play was when COVID vaccination drives were underway in the state. RRT volunteers helped community members register online for COVID vaccination, a process that had many steps and was unfamiliar to some.

## Impact of community formations

Both LSG and health department stakeholders reported meaningful engagement between communities and the health department, which they felt had a positive impact during the worst of the COVID outbreak in the state. But this was just the beginning, as pointed out by an outreach worker; there likely were future benefits as well:

> *We used to manage all those roles alone (COVID control activities) but with the introduction of RRT and social involvement, our burden was reduced substantially. . .So our roles were reduced because of the social responsibility initiative. One benefit because of the social responsibility initiative was that it made people aware of their roles and responsibilities. . . Now it is possible to tackle any other disease with this structure. That is the benefit.* (22_JHI)

PHC staff were able to cater the community members without any hiatus during the COVID- 19 pandemic. One of the Medical Officers at the PHCF attributed this feature to the volunteering efforts of RRTs.

> *Especially during COVID, they [RRTs] have been useful. Prior to the pandemic, such groups have been mostly inactive. . .But during the crisis we are able to utilise them in several ways. . .We deliver it [medicines] for COVID patients, their families as well as to those in the primary contact list [and to] the NCD patients as well. We do this with the help of our volunteers. Apart from the institution staff, we have so many volunteers including students.* (31_MO)

This optimism was not ubiquitous, however. Some participants raised concerns about the sustainability of RRTs as and when the pandemic restrictions were eased. An outreach worker in another district felt that "*RRT was formed during COVID time. . .since it was lockdown and people couldn't. . .go for their jobs, they were able to spend a lot of time in voluntary services. . .Later when the lockdown was lifted, people went back to their jobs.*" (11_HI)

Others in the same district felt that even if people go back to work, there may still be a minority–about one in ten volunteers, who may remain available and be able to help in the future.

## Discussion

Our study sought to identify various arrangements for community formations in the state for COVID-19 management and mitigation activities, the activities undertaken by them, and the impact these community formations had. In our study we learnt that there were myriad community formations which were existing before COVID-19 pandemic and hybrid models in which the existing volunteer groups were merged with newly formed ones like RRTs and there were context specific novel initiatives for engaging community in disaster/pandemic responses. Some even had a significant impact leading to its scale up across state like *Mash* program. Although the volunteering itself was not tied to political affiliations, there were instances of political affiliations organising youth to participate and volunteer. Community formations were recognised, trained, assigned tasks, supervised, and supported by LSG members and health system staff. These community volunteers in turn were seen as a big support for the health system actors as they were additional support for physical help to the health system and the LSG for the tasks they could not accomplish alone.

In the state, contextually relevant adaptations were carried out to create a pool of community volunteers who were working hard during the pandemic spike time in the state. Many countries have such a system in place with adaptations to their socio-cultural context. Low and lower-middle income countries have introduced a range of community participation interventions and community mobilizations to prevent communicable diseases. A systematic review revealed that prevalence of TB, Malaria, HIV/STIs were reduced due to community engagement along with reductions in indicators like maternal and neonatal mortality [30]. Evidence from communicable disease responses in Sub-Saharan Africa suggest that community participation helped in building confidence among the people for better containment of Ebola and COVID [2, 31, 32].

The coming together of various community formations and groups with government recognition is also significant. Kerala's creche workers were formally tasked with providing psycho social support, quarantine support, and linking populations to government systems where required [33]. Community kitchens at a mass scale were established with an aim to provide food for the quarantined, with a mission to guarantee everybody is free of hunger during lockdown [34, 35]. This initiative received recognition across the country with national leaders demanding replication in other states [36]. During this time there were departments like Health, LSG, the state police force, different state schemes like Kudumbashree, Integrated Child Development Services (a Government of India program that provides nutritional meals, preschool education, immunization, health check-up and referral services to children under six years of age and their mothers) that worked hand-in-hand to channelize various efforts for mitigating COVID-19 and the various effects like, unemployment, poverty, hunger and more, which demonstrated that the interdepartmental coordination was crucial to create Kerala model of COVID mitigation [37, 38]. India's neighbour, Bhutan also has experiences of whole-of-society approaches, involving multi-sectoral coordination between government ministries, private sector, local authorities, and civil society organizations to ensure service continuity during COVID [39]. Whole-of-society approaches have been promoted widely in both developing and developed nations, particularly in emergency management of pandemic or disaster mitigation activities [40–43]. This literature points to the synergistic relationship between multi-sectoral action and social participation in health in fronting an effective disaster response.

Given the changing scenario during COVID, the presence of conventional and formal volunteer groups did not restrict other community led informal or youth groups from joining efforts. Our study has found that some of the youth groups were affiliated with prominent

political parties. Our findings resonated with report from Kerala pointed to the efforts carried out by other informal groups for COVID mitigation efforts with some of them affiliated with political parties while others not [44]. There were number of student political organizations contributing towards pandemic response at community level [45–47], and RRTs routinely received support from political organizations. Another study from India showed that when there was a disconnect between people and health systems during a crisis, informal individual collectives and civil society organizations were mobilized to help people [48]. The situation in Kerala was different. While electoral politics may have motivated political party affiliated youth volunteers' participation (elections were being held around the same time), this is not evident in our study, nor in the literature we reviewed.

Our study participants noted the strong support provided by LSGs and health facility staff to RRT members to integrate their work with state COVID management processes. This was implemented through online/offline trainings provided by LSGs and further followed up by them. The volunteers were also given essential PPE through the State Disaster Response Fund, but no other benefits–in-kind or otherwise- were made available to them. Various studies have noted efforts to strengthen capacity of community volunteers and networks for effective community participation which included involving them in designing, planning and implementation of activities [5, 49]. It is noteworthy that this was <u>not</u> the case in our study in Kerala. Communities were not involved with design or planning of activities, but rather only implementation of activities. Elsewhere, wider involvement of community members has resulted in the formulation of unique local solutions [49]. The role of RRTs and other community volunteers in supporting (COVID and non-COVID) continuity of care has also already been noted, [50] although their role in actual planning was quite limited. Instead, the roles played were more aligned with findings from Oman where volunteers played three roles during COVID: health educators, data collectors, and social mobilizers [32]. COVID-19 created invited spaces and specified roles for citizens to be involved in voluntary and active community participation, and assist in the discharge of health system functions [51]. A study among African-Americans has pointed out the need for policy formulation for community participation. Such policy formulation would help develop community leadership to achieve health equity. This would also prepare the community members to act proactively in case of future crisis [52].

The broader context of decentralisation reforms in Kerala over the years [53], alongside various such efforts in and preceding the COVID context [37] appear to have created ample invited spaces for communities to contribute towards crisis response. A rapid evidence review of outbreaks of epidemics since 2000 reflected that the community participation should be a continuous and collaborative process, as the community members themselves are the most legitimate and influential persons in their respective communities [49]. The citizenship literature advocates for greater autonomy and proactive roles for community actors, which may include processes like citizen-sourcing, co-production of policy, data and information collection, service co-production, policy problem identification and solution creation [51]. Such practices of citizenship were not evident in the Kerala context however.

Relatedly, while COVID community formation responses in Kerala are incremental, in that they build on earlier experiences with disaster, research is needed to determine whether or how such efforts endure over time. This is also an area of policy attention: how may such efforts be nurtured and kept dynamic and active over time? Other research we have carried out on decentralisation and health suggests that institutionalisation of local innovations helps with scale, but can also stymie local relevance and adaptiveness [12–14]. Growing momentum around social participation for health, particularly in the context of Universal Health Coverage suggests that this critical role, of communities, in planning and determining their health, is crucial [54]. However, few examples exist that monitor or assess community action over time,

its forms, its impacts and indeed, its contradictions. Kerala provides fertile ground for this type of research given the relationships between political (LSGs and political parties) and civil society (NGO sector) which have layers of complexity and contradiction (studied in sectors outside of health; see for example Ramkumar [55], Justino [56], and Heller [57, 58]). This kind of exploration–in both academia and policymaking related to health–is needed.

## Strengths and limitations

The core purpose of our qualitative analysis was not to document or catalogue community or voluntary action, so we may not have captured all the efforts that may have been underway in each district. Further, our sample was skewed to reflect health system actor perspectives more than community representatives (more specifically RRT members and volunteers This paper does not reflect on the experiences or perceptions of community members other than leaders, who in some cases were themselves identified by health system actors (and could reflect selection bias). An analysis drawing directly from the community could provide important insights on roles as well as impact. It is quite possible that they view their engagement in different terms, we have initiated such an analysis and look forward to learning what it reveals!

## Conclusion

The COVID-19 pandemic created opportunities for LSGs and health system actors to invite and organise community participation and volunteers across a range of COVID control and prevention activities. To some extent this allowed the health system to focus on non-COVID service continuity. The history of participatory local governance in Kerala has allowed for the creation of invited spaces for participation, even as the terms of engagement are not decided by communities, nor are they involved more deeply in planning and organising health policy or services. Community members thus become "surge" human resources for pandemic response with limited roles beyond crisis contexts. The implications this has for health and for governance warrant greater study and consideration.

## Supporting information

**S1 File. Description of the dataset.**
(DOCX)

**S2 File. Interview topic guide.**
(DOCX)

## Acknowledgments

We express our gratitude to our interview participants who shared their thoughtful reflections with us and for making time out from their busy schedule while the pandemic mitigation activities were ongoing. We are also grateful for the support of the Kerala Department of Health and Family Welfare as well as various Local Self Government Institutions in the state.

## Author Contributions

**Conceptualization:** Gloria Benny, Devaki Nambiar.

**Data curation:** Gloria Benny, Hari Sankar D., Jaison Joseph.

**Formal analysis:** Gloria Benny, Hari Sankar D., Jaison Joseph, Devaki Nambiar.

**Funding acquisition:** Devaki Nambiar.

**Validation:** Hari Sankar D.

**Writing – original draft:** Gloria Benny, Hari Sankar D., Devaki Nambiar.

**Writing – review & editing:** Gloria Benny, Hari Sankar D., Jaison Joseph, Surya Surendran, Devaki Nambiar.

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
