## [Decision Letter · Decision Letter 0]

31 Jan 2023

PONE-D-22-29723On the forms, contributions and impacts of community mobilisation involved with Kerala’s COVID-19 response: Perspectives of health staff, local self government institution and community leadersPLOS ONE

Dear Dr. Benny,

Thank you for submitting your manuscript to PLOS ONE. After careful consideration, we feel that it has merit but does not fully meet PLOS ONE’s publication criteria as it currently stands. Therefore, we invite you to submit a revised version of the manuscript that addresses the points raised during the review process.

We look forward to receiving your revised manuscript.

Kind regards,

Ali B. Mahmoud, Ph.D.

Academic Editor

PLOS ONE

Journal Requirements:

"Wellcome Trust/DBT India Alliance Fellowship (https://www.indiaalliance.org) (Grant number IA/CPHI/16/1/502653) awarded to Dr. Devaki Nambiar. We express our gratitude to our interview participants who shared their thoughtful reflections with us and for making time out from their busy schedule while the pandemic mitigation activities were ongoing. We are also grateful for the support of the Kerala Department of Health and Family Welfare as well as various Local Self Government Institutions in Kerala"

"DN was awarded with Wellcome Trust/DBT India Alliance Fellowship (https://www.indiaalliance.org) (Grant number IA/CPHI/16/1/502653)

Reviewers' comments:

Reviewer's Responses to Questions

**Comments to the Author**

1. Is the manuscript technically sound, and do the data support the conclusions?

Reviewer #1: Yes

Reviewer #2: Yes

2. Has the statistical analysis been performed appropriately and rigorously? 

Reviewer #1: N/A

Reviewer #2: N/A

3. Have the authors made all data underlying the findings in their manuscript fully available?

Reviewer #1: Yes

Reviewer #2: No

4. Is the manuscript presented in an intelligible fashion and written in standard English?

Reviewer #1: Yes

Reviewer #2: Yes

5. Review Comments to the Author

Reviewer #1: In this study, the authors investigate the role of community participation in the response to the COVID-19 pandemic in Kerala, India, using interviews and thematic analysis. Results highlighted the formation and contributions of Rapid Response Teams (RRT). The manuscript is an interesting exploration of the interactions between volunteer groups and government-run public health. Questions and comments are below:

1. Background- aside from providing PPE, was any government funding or other resources provided to community groups working in the COVID-19 response? What was the source of this funding?

2. I was not able to access the interview questions as they were not included in the supplementary file. Were these questions piloted prior to starting interviewing and data collection?

3. How did the role of the RRTs evolve over the course of the pandemic?

4. How did the pandemic response and the role of community groups in Kerala compare to other states in India?

5. Most of the results and discussions focus on the beneficial role that the RRTs played, which is an important finding. However, I am interested in what respondents had to say about challenges and problems they experienced. Could these results be added?

Minor comment: Please write out all abbreviations upon first use (e.g. ASHA. LSGD) as many of these will not be familiar to readers.

Reviewer #2: Dear Editor,

I really appreciate the opportunity to review the manuscript PONE-D-22-29723 entitled:

"On the forms, contributions and impacts of community mobilisation involved with Kerala’s COVID-19 response: Perspectives of health staff, local self government institution and community leaders"

I commend the authors for describing this critical and timely issue. The paper is interesting and well-written; however, I would like to highlight some issues that merit revision:

The text appears very well written and interesting, but in some cases there are obvious typos or unclearly written sentences, such as, "and development and are trained in." In this regard, I ask the authors to carefully review the text of the manuscript.

I did not find several acronyms written in full, e.g. LSGD (LSG is found, but not LSGD), and so also

"HFWD";; I guess it is the same as DHFW, but in the text the acronyms should be clear and unicvoche for the reader, please, I ask the authors to evaluate and correct quesot aspetot.

Finally, in the conclusion, I found overenthusiastic tones. I fully understand that this is a good thing but writing "COVID-19 pandemic created opportunities for community participation to expand" exposes the paper to misinterpretation by uncaring journalists, who might titrate something like "covid brought good things." I ask the authors to mitigate the sentence with a small initial addition such as "although it brought harm to society and victims..."

6. PLOS authors have the option to publish the peer review history of their article (what does this mean?). If published, this will include your full peer review and any attached files.

Reviewer #1: No

Reviewer #2: **Yes: **Massimo Tusconi

---

## [Author Response · Author response to Decision Letter 0]

14 Apr 2023

I have addressed the comments and questions raised by both the reviewers and the same will be reflected in the revised manuscript.

---

## [Decision Letter · Decision Letter 1]

7 May 2023

On the forms, contributions and impacts of community mobilisation involved with Kerala’s COVID-19 response: Perspectives of health staff, local self government institution and community leaders

PONE-D-22-29723R1

Dear Dr. Benny,

We’re pleased to inform you that your manuscript has been judged scientifically suitable for publication and will be formally accepted for publication once it meets all outstanding technical requirements.

Kind regards,

Ali B. Mahmoud, Ph.D.

Academic Editor

PLOS ONE

Additional Editor Comments (optional):

Reviewers' comments:

Reviewer's Responses to Questions

**Comments to the Author**

1. If the authors have adequately addressed your comments raised in a previous round of review and you feel that this manuscript is now acceptable for publication, you may indicate that here to bypass the “Comments to the Author” section, enter your conflict of interest statement in the “Confidential to Editor” section, and submit your "Accept" recommendation.

Reviewer #1: All comments have been addressed

Reviewer #2: All comments have been addressed

2. Is the manuscript technically sound, and do the data support the conclusions?

Reviewer #1: Yes

Reviewer #2: Yes

3. Has the statistical analysis been performed appropriately and rigorously? 

Reviewer #1: N/A

Reviewer #2: Yes

4. Have the authors made all data underlying the findings in their manuscript fully available?

Reviewer #1: No

Reviewer #2: Yes

5. Is the manuscript presented in an intelligible fashion and written in standard English?

Reviewer #1: Yes

Reviewer #2: Yes

6. Review Comments to the Author

Reviewer #1: The revised manuscript has addressed the prior reviewer comments- thank you. I have no further comments.

Reviewer #2: Dear Editor,

I really appreciate the opportunity to review the manuscript PONE-D-22-29723-R1 entitled:

"On the forms, contributions and impacts of community mobilisation involved with Kerala’s COVID-19 response: Perspectives of health staff, local self government institution and community leaders"

The paper is very interesting and well-written, methodologically unexceptionable, and the new implementations provide a valid contribution to the work. Every requested correction has been done, and the manuscript is now suitable for publication

7. PLOS authors have the option to publish the peer review history of their article (what does this mean?). If published, this will include your full peer review and any attached files.

Reviewer #1: No

Reviewer #2: **Yes: **Massimo Tusconi

---

## [Editor Report · Acceptance letter]

29 May 2023

PONE-D-22-29723R1 

On the forms, contributions and impacts of community mobilisation involved with Kerala’s COVID-19 response: Perspectives of health staff, local self government institution and community leaders 

Dear Dr. Benny:

I'm pleased to inform you that your manuscript has been deemed suitable for publication in PLOS ONE. Congratulations! Your manuscript is now with our production department. 

Kind regards, 

on behalf of

Dr. Ali B. Mahmoud 

Academic Editor

PLOS ONE